# Robust Multimodal Graph Matching: Sparse Coding Meets Graph Matching

**Marcelo Fiori**
Universidad de la
República, Uruguay
mfiori@fing.edu.uy

**Pablo Sprechmann**
Duke University
Durham, NC 27708
pablo.sprechmann@duke.edu

**Joshua Vogelstein**
Duke University
Durham, NC 27708
jovo@math.duke.edu

**Pablo Musé**
Universidad de la
República, Uruguay
pmuse@fing.edu.uy

**Guillermo Sapiro**
Duke University
Durham, NC 27708
guillermo.sapiro@duke.edu

## Abstract

Graph matching is a challenging problem with very important applications in a wide range of fields, from image and video analysis to biological and biomedical problems. We propose a robust graph matching algorithm inspired in sparsity-related techniques. We cast the problem, resembling group or collaborative sparsity formulations, as a non-smooth convex optimization problem that can be efficiently solved using augmented Lagrangian techniques. The method can deal with weighted or unweighted graphs, as well as multimodal data, where different graphs represent different types of data. The proposed approach is also naturally integrated with collaborative graph inference techniques, solving general network inference problems where the observed variables, possibly coming from different modalities, are not in correspondence. The algorithm is tested and compared with state-of-the-art graph matching techniques in both synthetic and real graphs. We also present results on multimodal graphs and applications to collaborative inference of brain connectivity from alignment-free functional magnetic resonance imaging (fMRI) data. The code is publicly available.

## 1 Introduction

Problems related to graph isomorphisms have been an important and enjoyable challenge for the scientific community for a long time. The graph isomorphism problem itself consists in determining whether two given graphs are isomorphic or not, that is, if there exists an edge preserving bijection between the vertex sets of the graphs. This problem is also very interesting from the computational complexity point of view, since its complexity level is still unsolved: it is one of the few problems in NP not yet classified as P nor NP-complete (Conte et al., 2004). The graph isomorphism problem is contained in the (harder) graph matching problem, which consists in finding the exact isomorphism between two graphs. Graph matching is therefore a very challenging problem which has several applications, e.g., in the pattern recognition and computer vision areas. In this paper we address the problem of (potentially multimodal) graph matching when the graphs are not exactly isomorphic. This is by far the most common scenario in real applications, since the graphs to be compared are the result of a measuring or description process, which is naturally affected by noise.

Given two graphs $G_A$ and $G_B$ with $p$ vertices, which we will characterize in terms of their $p \times p$ adjacency matrices $\mathbf{A}$ and $\mathbf{B}$, the graph matching problem consists in finding a correspondence between the nodes of $G_A$ and $G_B$ minimizing some matching error. In terms of the adjacency matrices, this corresponds to finding a matrix $\mathbf{P}$ in the set of permutation matrices $\mathcal{P}$, such that it minimizes some distance between $\mathbf{A}$ and $\mathbf{P}\mathbf{B}\mathbf{P}^{\mathbf{T}}$. A common choice is the Frobenius norm $||\mathbf{A} - \mathbf{P}\mathbf{B}\mathbf{P}^{\mathbf{T}}||_F^2$, where $||\mathbf{M}||_F^2 = \sum_{ij} \mathbf{M}_{ij}^2$. The graph matching problem can be then stated as

$$\min_{\mathbf{P} \in \mathcal{P}} ||\mathbf{A} - \mathbf{P}\mathbf{B}\mathbf{P}^{\mathbf{T}}||_F^2 = \min_{\mathbf{P} \in \mathcal{P}} ||\mathbf{A}\mathbf{P} - \mathbf{P}\mathbf{B}||_F^2. \tag{1}$$

The combinatorial nature of the permutation search makes this problem NP in general, although polynomial algorithms have been developed for a few special types of graphs, like trees or planar graphs for example (Conte et al., 2004).

There are several and diverse techniques addressing the graph matching problem, including spectral methods (Umeyama, 1988) and problem relaxations (Zaslavskiy et al., 2009; Vogelstein et al., 2012; Almohamad & Duffuaa, 1993). A good review of the most common approaches can be found in Conte et al. (2004). In this paper we focus on the relaxation techniques for solving an approximate version of the problem. Maybe the simplest one is to relax the feasible set (the permutation matrices) to its convex hull, the set of doubly stochastic matrices $\mathcal{D}$, which consist of the matrices with non-negative entries such that each row and column sum up one: $\mathcal{D} = \{\mathbf{M} \in \mathbb{R}^{p \times p} : \mathbf{M}_{ij} \geq 0, \mathbf{M1} = \mathbf{1}, \mathbf{M}^T \mathbf{1} = \mathbf{1}\}$, $\mathbf{1}$ being the $p$-dimensional vector of ones. The relaxed version of the problem is

$$\hat{\mathbf{P}} = \arg \min_{\mathbf{P} \in \mathcal{D}} ||\mathbf{AP} - \mathbf{PB}||_F^2,$$

which is a convex problem, though the result is a doubly stochastic matrix instead of a permutation. The final node correspondence is obtained as the closest permutation matrix to $\hat{\mathbf{P}}$: $\mathbf{P}^* = \arg \min_{\mathbf{P} \in \mathcal{P}} ||\mathbf{P} - \hat{\mathbf{P}}||_F^2$, which is a linear assignment problem that can be solved in $O(p^3)$ by the Hungarian algorithm (Kuhn, 1955). However, this last step lacks any guarantee about the graph matching problem itself. This approach will be referred to as QCP for *quadratic convex problem*.

One of the newest approximate methods is the PATH algorithm by Zaslavskiy et al. (2009), which combines this convex relaxation with a concave relaxation. Another new technique is the FAQ method by Vogelstein et al. (2012), which solves a relaxed version of the Quadratic Assignment Problem. We compare the method here proposed to all these techniques in the experimental section.

The main contributions of this work are two-fold. Firstly, we propose a new and versatile formulation for the graph matching problem which is more robust to noise and can naturally manage multimodal data. The technique, which we call GLAG for Group lasso graph matching, is inspired by the recent works on sparse modeling, and in particular group and collaborative sparse coding. We present several experimental evaluations to back up these claims. Secondly, this proposed formulation fits very naturally into the alignment-free collaborative network inference problem, where we collaborative exploit non-aligned (possibly multimodal) data to infer the underlying common network, making this application never addressed before to the best of our knowledge. We assess this with experiments using real fMRI data.

The rest of this paper is organized as follows. In Section 2 we present the proposed graph matching formulation, and we show how to solve the optimization problem in Section 3. The joint collaborative network and permutation learning application is described in Section 4. Experimental results are presented in Section 5, and we conclude in Section 6.

## 2   Graph matching formulation

We consider the problem of matching two graphs that are not necessarily perfectly isomorphic. We will assume the following model: Assume that we have a noise free graph characterized by an adjacency matrix $\mathbf{T}$. Then we want to match two graphs with adjacency matrices $\mathbf{A} = \mathbf{T} + \mathbf{O_A}$ and $\mathbf{B} = \mathbf{P_o^T T P_o} + \mathbf{O_B}$, where $\mathbf{O_A}$ and $\mathbf{O_B}$ have a sparse number of non-zero elements of arbitrary magnitude. This realistic model is often used in experimental settings, e.g., (Zaslavskiy et al., 2009).

In this context, the QCP formulation tends to find a doubly stochastic matrix $\mathbf{P}$ which minimizes the "average error" between $\mathbf{AP}$ and $\mathbf{PB}$. However, these spurious mismatching edges can be thought of as outliers, so we would want a metric promoting that $\mathbf{AP}$ and $\mathbf{PB}$ share the same active set (non zero entries representing edges), with the exception of some sparse entries. This can be formulated in terms of the group Lasso penalization (Yuan & Lin, 2006). In short, the group Lasso takes a set of groups of coefficients and promotes that only some of these groups are active, while the others remain zero. Moreover, the usual behavior is that when a group is active, all the coefficients in the group are non-zero. In this particular graph matching application, we form $p^2$ groups, one per matrix entry $(i, j)$, each one consisting of the 2-dimensional vector $\big((\mathbf{AP})_{ij}, (\mathbf{PB})_{ij}\big)$. The proposed cost function is then the sum of the $l_2$ norms of the groups:

$$f(P) = \sum_{i,j} \big|\big|\big((\mathbf{AP})_{ij}, (\mathbf{PB})_{ij}\big)\big|\big|_2 . \tag{2}$$

Ideally we would like to solve the graph matching problem by finding the minimum of $f$ over the set of permutation matrices $\mathcal{P}$. Of course this formulation is still computationally intractable, so we solve the relaxed version, changing $\mathcal{P}$ by its convex hull $\mathcal{D}$, resulting in the convex problem

$$\tilde{\mathbf{P}} = \arg\min_{\mathbf{P}\in\mathcal{D}} f(\mathbf{P}). \tag{3}$$

As with the Frobenius formulation, the final step simply finds the closest permutation matrix to $\tilde{\mathbf{P}}$.

Let us analyze the case when $\mathbf{A}$ and $\mathbf{B}$ are the adjacency matrices of two isomorphic undirected unweighted graphs with $e$ edges and no self-loops. Since the graphs are isomorphic, there exist a permutation matrix $\mathbf{P_o}$ such that $\mathbf{A} = \mathbf{P_o}\mathbf{B}\mathbf{P_o^T}$.

**Lemma 1** *Under the conditions stated above, the minimum value of the optimization problem* (3) *is $2\sqrt{2}e$ and it is reached by $\mathbf{P_o}$, although the solution is not unique in general. Moreover, any solution $\mathbf{P}$ of problem* (3) *satisfies $\mathbf{AP} = \mathbf{PB}$.*

*Proof:* Let $(a)_k$ denote all the $p^2$ entries of $\mathbf{AP}$, and $(b)_k$ all the entries of $\mathbf{PB}$. Then $f(\mathbf{P})$ can be re-written as $f(\mathbf{P}) = \sum_k \sqrt{a_k^2 + b_k^2}$ .

Observing that $\sqrt{a^2 + b^2} \geq \frac{\sqrt{2}}{2}(a+b)$, we have

$$f(P) = \sum_k \sqrt{a_k^2 + b_k^2} \geq \sum_k \frac{\sqrt{2}}{2}(a_k + b_k) . \tag{4}$$

Now, since $\mathbf{P}$ is doubly stochastic, the sum of all the entries of $\mathbf{AP}$ is equal to the sum of all the entries of $\mathbf{A}$, which is two times the number of edges. Therefore $\sum_k a_k = \sum_k b_k = 2e$ and $f(\mathbf{P}) \geq 2\sqrt{2}e$.

The equality in (4) holds if and only if $a_k = b_k$ for all $k$, which means that $\mathbf{AP} = \mathbf{PB}$. In particular, this is true for the permutation $\mathbf{P_o}$, which completes the proof of all the statements. $\quad\square$

This Lemma shows that the fact that the weights in $\mathbf{A}$ and $\mathbf{B}$ are not compared in magnitude does not affect the matching performance when the two graphs are isomorphic and have equal weights. On the other hand, this property places a fundamental role when moving away from this setting. Indeed, since the group lasso tends to set complete groups to zero, and the actual value of the non-zero coefficients is less important, this allows to group very dissimilar coefficients together, if that would result in fewer active groups. This is even more evident when using the $l_\infty$ norm instead of the $l_2$ norm of the groups, and the optimization remains very similar to the one presented below. Moreover, the formulation remains valid when both graphs come from different modalities, a fundamental property when for example addressing alignment-free collaborative graph inference as presented in Section 4 (the elegance with which this graph matching formulation fits into such problem will be further stressed there). In contrast, the Frobenious-based approaches mentioned in the introduction are very susceptible to differences in edge magnitudes and not appropriate for multimodal matching[1].

## 3 Optimization

The proposed minimization problem (3) is convex but non-differentiable. Here we use an efficient variant of the Alternating Direction Method of Multipliers (ADMM) (Bertsekas & Tsitsiklis, 1989). The idea is to write the optimization problem as an equivalent artificially constrained problem, using two new variables $\boldsymbol{\alpha}, \boldsymbol{\beta} \in \mathbb{R}^{p\times p}$:

$$\min_{\mathbf{P}\in\mathcal{D}} \sum_{i,j} ||(\boldsymbol{\alpha}_{ij}, \boldsymbol{\beta}_{ij})||_2 \qquad s.t. \quad \boldsymbol{\alpha} = \mathbf{AP}, \quad \boldsymbol{\beta} = \mathbf{PB}. \tag{5}$$

The ADMOM method generates a sequence which converges to the minimum of the augmented Lagrangian of the problem:

$$L(\mathbf{P}, \boldsymbol{\alpha}, \boldsymbol{\beta}, \mathbf{U}, \mathbf{V}) = \sum_{i,j} ||(\boldsymbol{\alpha}_{ij}, \boldsymbol{\beta}_{ij})||_2 + \frac{c}{2}||\boldsymbol{\alpha} - \mathbf{AP} + \mathbf{U}||^2 + \frac{c}{2}||\boldsymbol{\beta} - \mathbf{PB} + \mathbf{V}||^2,$$

where $\mathbf{U}$ and $\mathbf{V}$ are related to the Lagrange multipliers and $c$ is a fixed constant.

The decoupling produced by the new artificial variables allows to update their values one at a time, minimizing the augmented Lagrangian $L$. We first update the pair $(\boldsymbol{\alpha}, \boldsymbol{\beta})$ while keeping fixed $(\mathbf{P}, \mathbf{U}, \mathbf{V})$; then we minimize for $\mathbf{P}$; and finally update $\mathbf{U}$ and $\mathbf{V}$, as described next in Algorithm 1.

---

**Input** : Adjacency matrices $\mathbf{A}, \mathbf{B}, c > 0$.
**Output**: Permutation matrix $\mathbf{P}^*$
Initialize $\mathbf{U} = \mathbf{0}, \mathbf{V} = \mathbf{0}, \mathbf{P} = \frac{1}{p}\mathbf{1}^T\mathbf{1}$
**while** *stopping criterion is not satisfied* **do**
$\quad (\boldsymbol{\alpha}^{t+1}, \boldsymbol{\beta}^{t+1}) = \arg\min_{\boldsymbol{\alpha},\boldsymbol{\beta}} \sum_{i,j} ||(\boldsymbol{\alpha}_{ij}, \boldsymbol{\beta}_{ij})||_2 + \frac{c}{2}||\boldsymbol{\alpha} - \mathbf{A}\mathbf{P}^t + \mathbf{U}^t||_F^2 + \frac{c}{2}||\boldsymbol{\beta} - \mathbf{P}^t\mathbf{B} + \mathbf{V}^t||_F^2$
$\quad \mathbf{P}^{t+1} = \arg\min_{\mathbf{P}\in\mathcal{D}} \ \frac{1}{2}||\boldsymbol{\alpha}^{t+1} - \mathbf{A}\mathbf{P} + \mathbf{U}^t||_F^2 + \frac{1}{2}||\boldsymbol{\beta}^{t+1} - \mathbf{P}\mathbf{B} + \mathbf{V}^t||_F^2$
$\quad \mathbf{U}^{t+1} = \mathbf{U}^t + \boldsymbol{\alpha}^{t+1} - \mathbf{A}\mathbf{P}^{t+1}$
$\quad \mathbf{V}^{t+1} = \mathbf{V}^t + \boldsymbol{\beta}^{t+1} - \mathbf{P}^{t+1}\mathbf{B}$
**end**
$\mathbf{P}^* = \arg\min_{\mathbf{Q}\in\mathcal{P}} ||\mathbf{Q} - \mathbf{P}||_F^2$

**Algorithm 1**: Robust graph matching algorithm. See text for implementation details of each step.

---

The first subproblem is decomposable into $p^2$ scalar problems (one for each matrix entry),

$$\min_{\boldsymbol{\alpha}_{ij}, \boldsymbol{\beta}_{ij}} ||(\boldsymbol{\alpha}_{ij}, \boldsymbol{\beta}_{ij})||_2 + \frac{c}{2}(\boldsymbol{\alpha}_{ij} - (\mathbf{A}\mathbf{P}^t)_{ij} + \mathbf{U}_{ij}^t)^2 + \frac{c}{2}(\boldsymbol{\beta}_{ij} - (\mathbf{P}^t\mathbf{B})_{ij} + \mathbf{V}_{ij}^t)^2.$$

From the optimality conditions on the subgradient of this subproblem, it can be seen that this can be solved in closed form by means of the well know vector soft-thresholding operator (Yuan & Lin, 2006): $S_v(\mathbf{b}, \lambda) = \left[1 - \frac{\lambda}{||\mathbf{b}||_2}\right]_+ \mathbf{b}$.

The second subproblem is a minimization of a convex differentiable function over a convex set, so general solvers can be chosen for this task. For instance, a projected gradient descent method can be used. However, this would require to compute several projections onto $\mathcal{D}$ per iteration, which is one of the computationally most expensive steps. Nevertheless, we can choose to solve a linearized version of the problem while keeping the convergence guarantees of the algorithm (Lin et al., 2011). In this case, the linear approximation of the first term is:

$$\frac{1}{2}||\boldsymbol{\alpha}^{t+1} - \mathbf{A}\mathbf{P} + \mathbf{U}^t||_F^2 \approx \frac{1}{2}||\boldsymbol{\alpha}^{t+1} - \mathbf{A}\mathbf{P}^k + \mathbf{U}^t||_F^2 + \langle \mathbf{g}^k, \mathbf{P} - \mathbf{P}^k \rangle + \frac{1}{2\tau}||\mathbf{P} - \mathbf{P}^k||_F^2,$$

where $\mathbf{g}^k = -\mathbf{A}^\mathbf{T}(\boldsymbol{\alpha}^{t+1} + \mathbf{U}^t - \mathbf{A}\mathbf{P}^k)$ is the gradient of the linearized term, $\langle\cdot,\cdot\rangle$ is the usual inner product of matrices, and $\tau$ is any constant such that $\tau < \frac{1}{\rho(\mathbf{A}^\mathbf{T}\mathbf{A})}$, with $\rho(\cdot)$ being the spectral norm.

The second term can be linearized analogously, so the minimization of the second step becomes

$$\min_{\mathbf{P}\in\mathcal{D}} \frac{1}{2}||\mathbf{P} - \underbrace{\left(\mathbf{P}^k + \tau\mathbf{A}^\mathbf{T}(\boldsymbol{\alpha}^{t+1} + \mathbf{U}^t - \mathbf{A}\mathbf{P}^k)\right)}_{\text{fixed matrix } \mathbf{C}}||_F^2 + \frac{1}{2}||\mathbf{P} - \underbrace{\left(\mathbf{P}^k + \tau(\boldsymbol{\beta}^{t+1} + \mathbf{V}^t - \mathbf{P}^k\mathbf{B})\mathbf{B}^\mathbf{T}\right)}_{\text{fixed matrix } \mathbf{D}}||_F^2$$

which is simply the projection of the matrix $\frac{1}{2}(\mathbf{C} + \mathbf{D})$ over $\mathcal{D}$.

Summarizing, each iteration consists of $p^2$ vector thresholdings when solving for $(\boldsymbol{\alpha}, \boldsymbol{\beta})$, one projection over $\mathcal{D}$ when solving for $\mathbf{P}$, and two matrix multiplications for the update of $\mathbf{U}$ and $\mathbf{V}$. The code is publicly available at `www.fing.edu.uy/~mfiori`.

# 4 Application to joint graph inference of not pre-aligned data

Estimating the inverse covariance matrix is a very active field of research. In particular the inference of the support of this matrix, since the non-zero entries have information about the conditional dependence between variables. In numerous applications, this matrix is known to be sparse, and in this regard the graphical Lasso has proven to be a good estimator for the inverse covariance matrix (Yuan & Lin, 2007; Fiori et al., 2012) (also for non-Gaussian data (Loh & Wainwright, 2012)). Assume that we have a $p$-dimensional multivariate normal distributed variable $X \sim \mathcal{N}(0, \Sigma)$; let $\mathbf{X} \in \mathbb{R}^{k\times p}$ be a data matrix containing $k$ independent observations of $X$, and $\mathbf{S}$ its empirical covariance matrix. The graphical Lasso estimator for $\mathbf{\Sigma}^{-1}$ is the matrix $\mathbf{\Theta}$ which solves the optimization problem

$$\min_{\mathbf{\Theta}\succ 0} \ \text{tr}(\mathbf{S}\mathbf{\Theta}) - \log\det\mathbf{\Theta} + \lambda\sum_{i,j}|\mathbf{\Theta}_{ij}|, \tag{6}$$

which corresponds to the maximum likelihood estimator for $\boldsymbol{\Sigma}^{-1}$ with an $l_1$ regularization.

Collaborative network inference has gained a lot of attention in the last years (Chiquet et al., 2011), specially with fMRI data, e.g., (Varoquaux et al., 2010). This problem consist of estimating two (or more) matrices $\boldsymbol{\Sigma}_A^{-1}$ and $\boldsymbol{\Sigma}_B^{-1}$ from data matrices $\mathbf{X}_A$ and $\mathbf{X}_B$ as above, with the additional prior information that the inverse covariance matrices share the same support. The joint estimation of $\boldsymbol{\Theta}^A$ and $\boldsymbol{\Theta}^B$ is performed by solving

$$\min_{\boldsymbol{\Theta}^A \succ 0, \boldsymbol{\Theta}^B \succ 0} \operatorname{tr}(\mathbf{S}^A \boldsymbol{\Theta}^A) - \log \det \boldsymbol{\Theta}^A + \operatorname{tr}(\mathbf{S}^B \boldsymbol{\Theta}^B) - \log \det \boldsymbol{\Theta}^B + \lambda \sum_{i,j} ||(\boldsymbol{\Theta}_{ij}^A, \boldsymbol{\Theta}_{ij}^B)||_2 , \quad (7)$$

where the first four terms correspond to the maximum likelihood estimators for $\boldsymbol{\Theta}^A, \boldsymbol{\Theta}^B$, and the last term is the group Lasso penalty which promotes that $\boldsymbol{\Theta}^A$ and $\boldsymbol{\Theta}^B$ have the same active set.

This formulation relies on the limiting underlying assumption that the variables in both datasets (the columns of $\mathbf{X}_A$ and $\mathbf{X}_B$) are in correspondence, i.e., the graphs determined by the adjacency matrices $\boldsymbol{\Theta}^A$ and $\boldsymbol{\Theta}^B$ are aligned. However, this is in general not the case in practice. Motivated by the formulation presented in Section 2, we propose to overcome this limitation by incorporating a permutation matrix into the optimization problem, and jointly learn it on the estimation process. The proposed optimization problem is then given by

$$\min_{\substack{\boldsymbol{\Theta}^A, \boldsymbol{\Theta}^B \succ 0 \\ \mathbf{P} \in \mathcal{P}}} \operatorname{tr}(\mathbf{S}^A \boldsymbol{\Theta}^A) - \log \det \boldsymbol{\Theta}^A + \operatorname{tr}(\mathbf{S}^B \boldsymbol{\Theta}^B) - \log \det \boldsymbol{\Theta}^B + \lambda \sum_{i,j} ||((\boldsymbol{\Theta}^A \mathbf{P})_{ij}, (\mathbf{P} \boldsymbol{\Theta}^B)_{ij})||_2.$$

$$(8)$$

Even after the relaxation of the constraint $\mathbf{P} \in \mathcal{P}$ to $\mathbf{P} \in \mathcal{D}$, the joint minimization of (8) over $(\boldsymbol{\Theta}^A, \boldsymbol{\Theta}^B)$ and $\mathbf{P}$ is a non-convex problem. However it is convex when minimized only over $(\boldsymbol{\Theta}^A, \boldsymbol{\Theta}^B)$ or $\mathbf{P}$ leaving the other fixed. Problem (8) can be then minimized using a block-coordinate descent type of approach, iteratively minimizing over $(\boldsymbol{\Theta}^A, \boldsymbol{\Theta}^B)$ and $\mathbf{P}$.

The first subproblem (solving (8) with $\mathbf{P}$ fixed) is a very simple variant of (7), which can be solved very efficiently by means of iterative thresholding algorithms (Fiori et al., 2013). In the second subproblem, since $(\boldsymbol{\Theta}^A, \boldsymbol{\Theta}^B)$ are fixed, the only term to minimize is the last one, which corresponds to the graph matching formulation presented in Section 2.

# 5 Experimental results

We now present the performance of our algorithm and compare it with the most recent techniques in several scenarios including synthetic and real graphs, multimodal data, and fMRI experiments. In the cases where there is a "ground truth," the performance is measured in terms of the *matching error*, defined as $||\mathbf{A}_o - \mathbf{P}\mathbf{B}_o\mathbf{P}^{\mathbf{T}}||_F^2$, where $\mathbf{P}$ is the obtained permutation matrix and $(\mathbf{A}_o, \mathbf{B}_o)$ are the original adjacency matrices.

## 5.1 Graph matching: Synthetic graphs

We focus here in the traditional graph matching problem of undirected weighted graphs, both with and without noise. More precisely, let $\mathbf{A}_o$ be the adjacency matrix of a random weighted graph and $\mathbf{B}_o$ a permuted version of it, generated with a random permutation matrix $\mathbf{P}_o$, i.e., $\mathbf{B}_o = \mathbf{P}_o^T \mathbf{A}_o \mathbf{P}_o$. We then add a certain number $N$ of random edges to $\mathbf{A}_o$ with the same weight distribution as the original weights, and another $N$ random edges to $\mathbf{B}_o$, and from these noisy versions we try to recover the original matching (or any matching between $\mathbf{A}_o$ and $\mathbf{B}_o$, since it may not be unique).

We show the results using three different techniques for the generation of the graphs: the Erdős-Rényi model (Erdős & Rényi, 1959), the model by Barabási & Albert (1999) for scale-free graphs, and graphs with a given degree distribution generated with the BTER algorithm (Seshadri et al., 2012). These models are representative of a wide range of real-world graphs (Newman, 2010). In the case of the BTER algorithm, the degree distribution was generated according to a geometric law, that is: $\operatorname{Prob}(\text{degree} = t) = (1 - e^{-\mu})e^{\mu t}$.

We compared the performance of our algorithm with the technique by Zaslavskiy et al. (2009) (referred to as PATH), the FAQ method described in Vogelstein et al. (2012), and the QCP approach.

Figure 1 shows the matching error as a function of the noise level for graphs with $p = 100$ nodes (top row), and for $p = 150$ nodes (bottom row). The number of edges varies between 200 and 400 for graphs with 100 nodes, and between 300 and 600 for graphs with 150 nodes, depending on the model. The performance is averaged over 100 runs. This figure shows that our method is more stable, and consistently outperforms the other methods (considered state-of-the-art), specially for noise levels in the low range (for large noise levels, is not clear what a "true" matching is, and in addition the sparsity hypothesis is no longer valid).

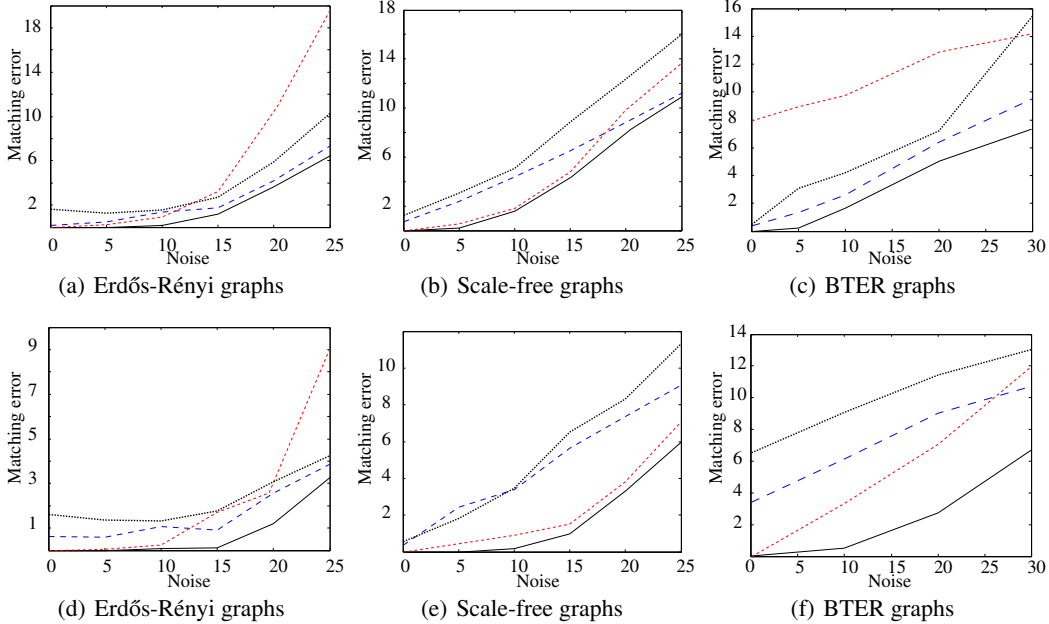

**Figure 1:** Matching error for synthetic graphs with $p = 100$ nodes (top row) and $p = 150$ nodes (bottom row). In solid black our proposed GLAG algorithm, in long-dashed blue the PATH algorithm, in short-dashed red the FAQ method, and in dotted black the QCP.

## 5.2 Graph matching: Real graphs

We now present similar experiments to those in the previous section but with real graphs. We use the *C. elegans* connectome. *Caenorhabditis elegans* is an extensively studied roundworm, whose somatic nervous system consists of 279 neurons that make synapses with other neurons. The two types of connections (chemical and electrical) between these 279 neurons have been mapped (Varshney et al., 2011), and their corresponding adjacency matrices, $\mathbf{A}_c$ and $\mathbf{A}_e$, are publicly available.

We match both the chemical and the electrical connection graphs against noisy artificially permuted versions of them. The permuted graphs are constructed following the same procedure used in Section 5.1 for synthetic graphs. The weights of the added noise follow the same distribution as the original weights. The results are shown in Figure 2. These results suggest that from the prior art, the PATH algorithm is more suitable for the electrical connection network, while the FAQ algorithm works better for the chemical one. Our method outperforms both of them for both types of connections.

## 5.3 Multimodal graph matching

One of the advantages of the proposed approach is its capability to deal with multimodal data. As discussed in Section 2, the group Lasso type of penalty promotes the supports of $\mathbf{AP}$ and $\mathbf{PB}$ to be identical, almost independently of the actual values of the entries. This allows to match weighted graphs where the weights may follow completely different probability distributions. This is commonly the case when dealing with multimodal data: when a network is measured using significantly different modalities, one expects the underlying connections to be the same but no relation can be assumed between the actual weights of these connections. This is even the case for example for fMRI data when measured with different instruments. In what follows, we evaluate the performance of the proposed method in two examples of multimodal graph matching.

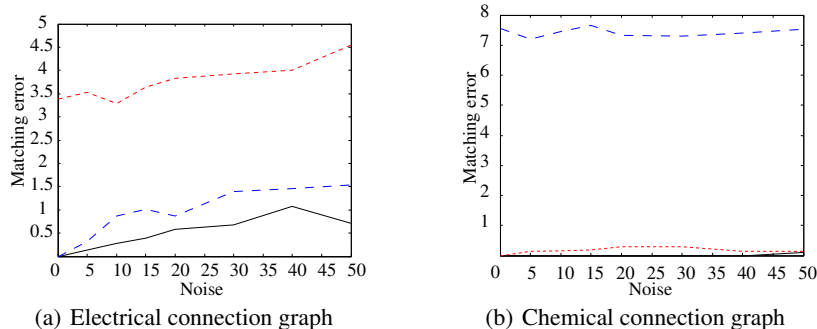

(a) Electrical connection graph       (b) Chemical connection graph

**Figure 2:** Matching error for the C. elegans connectome, averaged over 50 runs. In solid black our proposed GLAG algorithm, in long-dashed blue the PATH algorithm, and in short-dashed red the FAQ method. Note that in the chemical connection graph, the matching error of our algorithm is zero until noise levels of $\approx 50$.

We first generate an auxiliary binary random graph $\mathbf{A}_b$ and a permuted version $\mathbf{B}_b = \mathbf{P}_o^T \mathbf{A}_b \mathbf{P}_o$. Then, we assign weights to the graphs according to distributions $p_A$ and $p_B$ (that will be specified for each experiment), thus obtaining the weighted graphs $\mathbf{A}$ and $\mathbf{B}$. We then add noise consisting of spurious weighted edges following the same distribution as the original graphs (i.e., $p_A$ for $\mathbf{A}$ and $p_B$ for $\mathbf{B}$). Finally, we run all four graph matching methods to recover the permutation. The matching error is measured in the unweighted graphs as $||\mathbf{A}_b - \mathbf{P}\mathbf{B}_b\mathbf{P}^T||_F$. Note that while this metric might not be appropriate for the optimization stage when considering multimodal data, it is appropriate for the actual error evaluation, measuring mismatches. Comparing with the original permutation matrix may not be very informative since there is no guarantee that the matrix is unique, even for the original noise-free data.

Figures 3(a) and 3(b) show the comparison when the weights in both graphs are Gaussian distributed, but with different means and variances. Figures 3(c) and 3(d) show the performances when the weights of $\mathbf{A}$ are Gaussian distributed, and the ones of $\mathbf{B}$ follow a uniform distribution. See captions for details. These results confirm the intuition described above, showing that our method is more suitable for multimodal graphs, specially in the low range of noise.

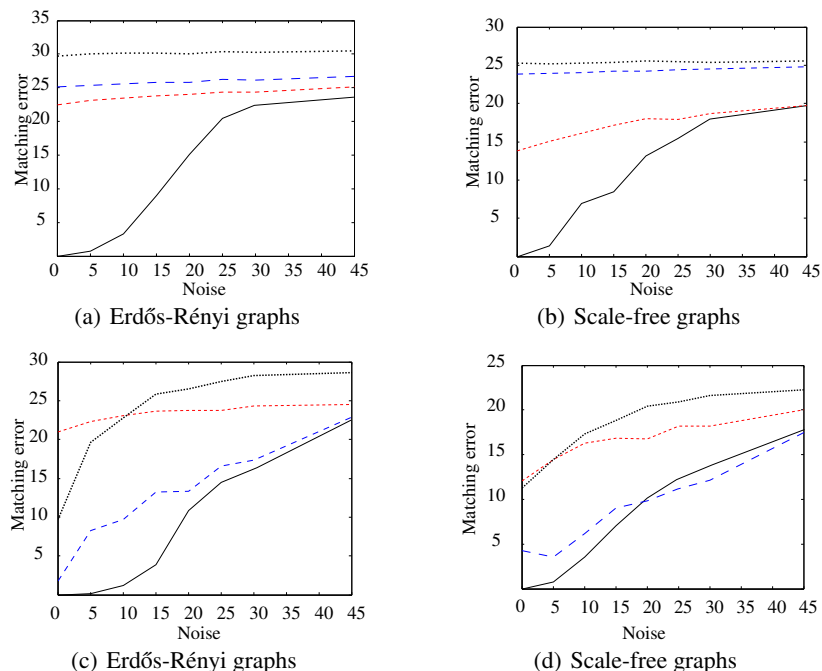

(a) Erdős-Rényi graphs       (b) Scale-free graphs

(c) Erdős-Rényi graphs       (d) Scale-free graphs

**Figure 3:** Matching error for multimodal graphs with $p = 100$ nodes. In (a) and (b), weights in $\mathbf{A}$ are $\mathcal{N}(1, 0.4)$ and weights in $\mathbf{B}$ are $\mathcal{N}(4, 1)$. In (c) and (d), weights in $\mathbf{A}$ are $\mathcal{N}(1, 0.4)$ and weights in $\mathbf{B}$ are uniform in $[1, 2]$. In solid black our proposed GLAG algorithm, in long-dashed blue the PATH algorithm, in short-dashed red the FAQ method, and in dotted black the QCP.

## 5.4 Collaborative inference

In this last experiment, we illustrate the application of the permuted collaborative graph inference presented in Section 4 with real resting-state fMRI data, publicly available (Nooner, 2012). We consider here test-retest studies, that is, the same subject undergoing resting-state fMRI in two different sessions separated by a break. Each session consists of almost 10 minutes of data, acquired with a sampling period of $0.645s$, producing about 900 samples per study. The CC200 atlas (Craddock et al., 2012) was used to extract the time-series for the $\approx 200$ regions of interest (ROIs), resulting in two data matrices $\mathbf{X}^A, \mathbf{X}^B \in \mathbb{R}^{900 \times 200}$, corresponding to test and retest respectively.

To illustrate the potential of the proposed framework, we show that using only part of the data in $\mathbf{X}^A$ and part of the data in a permuted version of $\mathbf{X}^B$, we are able to infer a connectivity matrix almost as accurately as using the whole data. Working with permuted data is very important in this application in order to handle possible miss-alignments to the atlas.

Since there is no ground truth for the connectivity, and as mentioned before the collaborative setting (7) has already been proven successful, we take as ground truth the result of the collaborative inference using the empirical covariance matrices of $\mathbf{X}^A$ and $\mathbf{X}^B$, denoted by $\mathbf{S}^A$ and $\mathbf{S}^B$. The result of this collaborative inference procedure are the two inverse covariance matrices $\mathbf{\Theta}^A_{GT}$ and $\mathbf{\Theta}^B_{GT}$. In short, the gold standard built for this experiment are found by solving (obtained with the entire data)

$$\min_{\mathbf{\Theta}^A \succ 0, \mathbf{\Theta}^B \succ 0} \text{tr}(\mathbf{S}^A \mathbf{\Theta}^A) - \log \det \mathbf{\Theta}^A + \text{tr}(\mathbf{S}^B \mathbf{\Theta}^B) - \log \det \mathbf{\Theta}^B + \lambda \sum_{i,j} ||(\mathbf{\Theta}^A_{ij}, \mathbf{\Theta}^B_{ij})||_2 \ .$$

Now, let $\mathbf{X}^A_H$ be the first 550 samples of $\mathbf{X}^A$, and $\mathbf{X}^B_H$ the first 550 samples of $\mathbf{X}^B$, which correspond to a little less than 6 minutes of study. We compute the empirical covariance matrices $\mathbf{S}^A_H$ and $\mathbf{S}^B_H$ of these data matrices, and we artificially permute the second one: $\tilde{\mathbf{S}}^B_H = \mathbf{P}^T_o \mathbf{S}^B_H \mathbf{P}_o$. With these two matrices $\mathbf{S}^A_H$ and $\tilde{\mathbf{S}}^B_H$ we run the algorithm described in Section 4, which alternately computes the inverse covariance matrices $\mathbf{\Theta}^A_H$ and $\mathbf{\Theta}^B_H$ and the matching $\mathbf{P}$ between them.

We compare this approach against the computation of the inverse covariance matrix using only one of the studies. Let $\mathbf{\Theta}^A_s$ and $\mathbf{\Theta}^B_s$ be the results of the graphical Lasso (6) using $\mathbf{S}^A$ and $\mathbf{S}^B$:

$$\mathbf{\Theta}^K_s = \underset{\mathbf{\Theta} \succ 0}{\text{argmin}} \ \ \text{tr}(\mathbf{S}^K \mathbf{\Theta}) - \log \det \mathbf{\Theta} + \lambda \sum_{i,j} |\mathbf{\Theta}_{ij}| \ , \quad \text{for } K = \{A, B\}.$$

This experiment is repeated for 5 subjects in the database. The errors $||\mathbf{\Theta}^A_{GT} - \mathbf{\Theta}^A_s||_F$ and $||\mathbf{\Theta}^A_{GT} - \mathbf{\Theta}^A_H||_F$ are shown in Figure 4. The errors for $\mathbf{\Theta}^B$ are very similar. Using less than 6 minutes of each study, with the variables not pre-aligned, the permuted collaborative inference procedure proposed in Section 4 outperforms the classical graphical Lasso using the full 10 minutes of study.

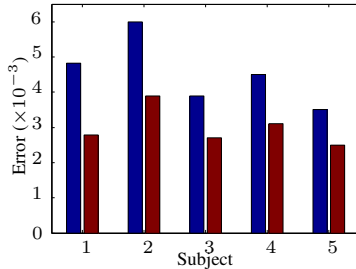

**Figure 4:** Inverse covariance matrix estimation for fMRI data. In blue, error using one complete 10 minutes study: $||\mathbf{\Theta}^A_{GT} - \mathbf{\Theta}^A_s||_F$. In red, error $||\mathbf{\Theta}^A_{GT} - \mathbf{\Theta}^A_H||_F$ with collaborative inference using about 6 minutes of each study, but solving for the unknown node permutations at the same time.

## 6 Conclusions

We have presented a new formulation for the graph matching problem, and proposed an optimization algorithm for minimizing the corresponding cost function. The reported results show its suitability for the graph matching problem of weighted graphs, outperforming previous state-of-the-art methods, both in synthetic and real graphs. Since in the problem formulation the weights of the graphs are not compared explicitly, the method can deal with multimodal data, outperforming the other compared methods. In addition, the proposed formulation naturally fits into the pre-alignment-free collaborative network inference framework, where the permutation is estimated together with the underlying common network, with promising preliminary results in applications with real data.

**Acknowledgements:** Work partially supported by ONR, NGA, NSF, ARO, AFOSR, and ANII.

## Footnotes

[1]If both graphs are binary and we limit to permutation matrices (for which there are no algorithms known to find the solution in polynomial time), then the minimizers of (2) and (1) are the same (Vince Lyzinski, personal communication).

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
