[Reviews · NeurIPS 2013]

Submitted by Assigned_Reviewer_4

This paper examines the problem of approximate graph matching (isomorphism). Given graphs G, H with p nodes, represented by respective adjacency matrices A, B,
Find a permutation matrix P that best ``matches’’ AP and PB.

The paper indicates that this is a problem with a long history. The fundamental problem is combinatorial and its complexity is not fully understood. This has led to various relations of the problem. For example to minimize a convex matching metric under the constraint that the permutation P is relaxed to a doubly stochastic matrix
**[reference needed at this point in the paper (line 061)].
Once the resulting convex problem is solved one then finds the nearest permutation matrix to the solution.

The authors point out that the traditional Frobenius norm matching metric
||AP-PB||_F^2, does not produce a common core shared graph with a sparse set of outlier edges. Hence they propose a group lasso approach in which the matching metric is the sum of the Euclidean norms of the corresponding entries of AP and PB (equation 2). This is motivated by the heuristic argument that the group lasso will tend to set most groups of coefficients to zero with only a few groups nonzero.
So the metric will seek a permutation that matrices many edges and will have a sparse set of edges that do not match well.

The only theoretical result of the paper is Lemma 1, which verifies that the proposed matching metric makes sense in the very special case of two isomorphic, undirected, unweighted graphs with no self loops (and obviously no noise in the form of randomly added edges). The lemma indicates that in this case, minimizing the metric will achieve AP=PB with P doubly stochastic. The proof is very elementary.

The authors then propose an optimization procedure for minimizing the new matching metric subject to P being doubly stochastic (problem (3)). This is based on a linearized variation of ADMM drawing heavily from the work by Lin et al (NIPS 2011).
**[Note: possible typo line 154: ADMOM ?].

Section 4, shows how the above ideas could be applied to collaborative inverse covariance estimation in the spirit of the graphical lasso. This leads to a non convex problem, but which is convex when either of two disjoint set of variable is fixed. This gives an alternating, two phase sequence of convex problems.
The basic problem seems to have its origins in Chiquet 2011, where it is addressed using a different method.

The experimental section of paper describes four experiments. The first explores performance of the proposed method and algorithm on several synthetic forms of graphs. The proposed method appears to have significant advantages in these experiments. The second experiment involves graph matching on real graphs from the connectome of C. elegans. There are two graphs (chemical connections and electrical connections). Although the data is real the problem solved is artificial: each graph is first permuted, edge noise is added and then this graph is matched to its original.

Experiment three explores the proposed method’s ability to match graphs with edge weights drawn from different distributions (a.k.a. multimodal graph matching). This is explored using synthetic datasets in the spirit of the first experiment. At low noise levels the proposed method seems to have a distinct advantage over competing methods – note however that in this case this means there is an (almost) exact match – this is unlikely to happen in real-world datasets.

The fourth experiment on collaborative inference is also based on real data – in this case resting state fMRI. However, the actual experiment is synthetic with the data to be matched obtained by permuting the real data. This is an interesting experiment. It shows that that the proposed method has potential advantages over the usual application of the graphical lasso – but is this surprising? The graphical lasso uses each graph separately, while the proposed method allows the graphs to collaborate.
This is obviously a good thing and should lead to improved results. The experiment confirms that this is the case.

Let me note that in several places the authors emphasize that the proposed method shows promising performance “on real data”. For example, phrases of this type appear twice in the conclusion. This is a little misleading since although the data is obtained from real experiments the actual tests performed are synthetic and hence are not reflective of actual alignments that would occur in practice. It would be better to say: “shows promising performance using real data to form synthetic alignment problems”.

Summary: Overall assessment: this is a well-written paper with some interesting practical ideas. There is not much new in the way of theoretical development and the work appears to draw heavily from the recent work of others - but the algorithm produced shows promise. The experiments are interesting – if somewhat artificial. The paper is suitable for presentation at NIPS and should be of interest to a wide audience.

Submitted by Assigned_Reviewer_5

This paper poses the multimodal graph matching problem as a convex optimization problem, and solves it using augmented Langrangian techniques (viz., ADMM). This is an important problem with application in several fields. Experimental results on synthetic and multiple real world datasets demonstrate effectiveness of the proposed approach. The paper is quite well written and is easy to follow. Some suggestions for improvements are listed below.

I wonder whether the author considered L1 norm for the group terms in Eqn (2)?

It might be informative to include some (empirical) convergence details of Algorithm 1.

In fig 2, matching error differences among different methods don't seem significant. Some clarification will help.

In all the experiments, the datasets used are rather small, usually involving <= 200 nodes in the graph. Is this due to scalability limitations of the proposed approach? Some details on the computational complexity of the proposed approach (and also runtimws) will be helpful.

Normalizing (aligning) brains of *different* subjets participating in the same study is an important preprocessing step in any fMRI data analysis. This is another graph matching problem whether the proposed method might be interesting to apply (just a suggestion for future work, not for this paper).

Typo:
Lines 157: ADOMM => ADMM
Summary: This paper poses the multimodal graph matching problem as a convex optimization problem, and solves it using an augmented Lagrangian approach. Experimental results demonstrate effectiveness of the proposed method. Graph matching is an important problem and the proposed approach should be of interest to the NIPS community.

Submitted by Assigned_Reviewer_6

The paper presents an algorithm for approximate graph matching. The idea is very simple, still quite effective: rather than modeling the mismatch between AP and PB (P permutation matrix) as the sum of squared errors, it uses the group-lasso cost function, forcing the edges in the two permuted matrices to be either both present or both absent (and keeping the number of edges small). The objective function is then applied to the problem of graph inference.

The paper is well presented and the proposed approach is clearly motivated and explained.

Experimental results are given for artificial data, for the C. elegans connectome, and for fMRI data. The latter set of experiments is potentially the most interesting one. However, from a neuroscience point of view, results seem to be auto-referential, lacking a convincing validation via some form of biological/neurological evidence. This is of course not easy to obtain and for a computational venue like NIPS perhaps should not be a must (the methodology is interesting per se and results on other data sets are convincing).
Summary: Interesting and well presented paper. Experimental results could be more interesting if supported by some form of evidence.
Author Feedback

Author rebuttal: First of all we want to thank the reviewers for their deep and careful revision of the paper, and their very constructive comments.
All their suggestions will be carefully addressed in the camera-ready/revision of the paper. Below we just address
the main issues (though all reviewers reported only minor issues).

Rev. #1 (Assigned_Reviewer_4)
======

We will add the requested reference to the relaxation of the Frobenius norm optimization.

We will add supporting references regarding the theoretical guarantees of group Lasso, which are not just heuristics, to produce joint active sets.

While the proof of Lemma 1 is given for undirected graphs with no self-loops, the result holds for directed graphs and with self-loops. We chose to include this reduced version in order to keep the proof simple, and also because of space restrictions. However, the proof is very similar.

We will discuss more the noise issue for the multimodal experiment; note that at high noise, the actual matching is lost and then the
real relevance of matching the graphs is doubtful. In other words, the validity of the match at high noise is not clear, since the graphs are not
matchable/compatible any more. We will discuss this further in the revision.

The goal of the fMRI experiment was indeed to further validate the approach, we will discuss this further and illustrate also how this technique helps in classification, meaning classifying for example gender from the network graph.

We will clarify and correct the issue with "real data" for the experiments.

Rev. #2 (Assigned_Reviewer_5)
======

An L1 norm inside the group term in (2), due to the separability of the norm, will not promote group sparsity but
general sparsity on the matrix itself. Therefore, the optimization would force both matrices AP and PB to be sparse, but with no link between their support. This will be made clear in the revised version

We will add convergence details (not only empirical but also theoretical) on Algorithm 1.

We will further discuss the results in Fig. 2, which we believe show significant improvement, in particular because the proposed method outperforms state-of-the-art in all tested datasets, sometimes by a little and sometimes by a lot. Previous approaches were better for some datasets and worse for others, while ours is always better; we will add the average improvement to further stress this.

The size of the real world graphs were fixed by the application of course. As for the synthetic examples, we tried to show results for graphs of the same order as in the state-of-the-art paper by Zaslavskiy et. al. However, the approach is scalable indeed. We will discuss the computational complexity, which is derived from the state-of-the-art optimization algorithms used.

Thanks for the suggestion on fMRI, we are now seriously pursuing this line of research with large datasets collected at our and other universities and will present some of the results at the conference.

Rev. #3 (Assigned_Reviewer_6)
======

While the fMRI experiment was presented to further stress and confirm the validity of the proposed framework, we are currently extending
this and will present some of the findings at the conference. In particular, this type of approach is significantly improving our performance in network classification, e.g., how to classify brains according to the computed activity network. We are observing double digit improvements in performance and this will be reported.